# Generating functional protein variants with variational autoencoders

**Alex Hawkins-Hooker, Florence Depardieu, Sebastien Baur, Guillaume Couairon, Arthur Chen, David Bikard** *

Synthetic Biology Group, Microbiology Department, Institut Pasteur, Paris, France

* david.bikard@pasteur.fr

## Abstract

The vast expansion of protein sequence databases provides an opportunity for new protein design approaches which seek to learn the sequence-function relationship directly from natural sequence variation. Deep generative models trained on protein sequence data have been shown to learn biologically meaningful representations helpful for a variety of downstream tasks, but their potential for direct use in the design of novel proteins remains largely unexplored. Here we show that variational autoencoders trained on a dataset of almost 70000 luciferase-like oxidoreductases can be used to generate novel, functional variants of the *luxA* bacterial luciferase. We propose separate VAE models to work with aligned sequence input (MSA VAE) and raw sequence input (AR-VAE), and offer evidence that while both are able to reproduce patterns of amino acid usage characteristic of the family, the MSA VAE is better able to capture long-distance dependencies reflecting the influence of 3D structure. To confirm the practical utility of the models, we used them to generate variants of *luxA* whose luminescence activity was validated experimentally. We further showed that conditional variants of both models could be used to increase the solubility of *luxA* without disrupting function. Altogether 6/12 of the variants generated using the unconditional AR-VAE and 9/11 generated using the unconditional MSA VAE retained measurable luminescence, together with all 23 of the less distant variants generated by conditional versions of the models; the most distant functional variant contained 35 differences relative to the nearest training set sequence. These results demonstrate the feasibility of using deep generative models to explore the space of possible protein sequences and generate useful variants, providing a method complementary to rational design and directed evolution approaches.

## Author summary

The design of novel proteins with specified function and biochemical properties is a long-standing goal in bio-engineering with applications across medicine and nanotechnology. Despite the impressive achievements of traditional approaches, a great deal of scope remains for the development of data-driven methods capable of exploiting the record of natural sequence variation available in protein databases. Deep generative models such as

**Data Availability Statement:** Training data were obtained from InterPro (IPR011251). Data relating to experimental characterisation of generated sequences are within the manuscript and its Supporting information files. Python

implementations of models and training procedure are available at https://github.com/alex-hh/deep-protein-generation.

**Funding:** This work was supported by the French Government's Investissement d'Avenir program and by Laboratoire d'Excellence 'Integrative Biology of Emerging Infectious Diseases' (ANR-10-LABX-62-IBEID) to D.B. The funders had no role in study design, data collection and analysis, decision to publish, or preparation of the manuscript.

**Competing interests:** The authors have declared that no competing interests exist.

variational autoencoders (VAEs) have shown remarkable success in synthesising realistic data samples across a range of modalities, driving recent interest in developing such models for proteins. However, experimental evidence for the viability of such techniques in practical protein design settings remains scarce. Here we show that VAEs trained on the family of luciferase-like oxidoreductases can be used to generate functional variants of the *luxA* bacterial luciferase. We compare the use of raw and aligned sequences as input to the model, providing evidence that models trained on aligned data are better able to learn functional constraints. Finally, we demonstrate the possibility of controlling desired properties of the designed sequences, by using conditional versions of the VAE models to increase the solubility of the wild-type *luxA* sequence from *P. luminescens*.

## Introduction

Recombinant proteins have found uses in many medical and industrial applications where it is frequently desirable to identify protein variants with modified properties such as improved stability, catalytic activity, and modified substrate preferences. The systematic exploration of protein variants is made extremely challenging by the enormous space of possible sequences and the difficulty of accurately predicting protein fold and function. Directed evolution approaches enable a more or less random local search of sequence space but are typically limited to the exploration of sequences differing by only a few mutations from a given natural sequence [1, 2]. When knowledge of the protein structure is available, computer aided rational design can help identify interesting modifications [3]. Beyond the identification of sequence variants, computational approaches have enabled the generation of small synthetic protein domains that mimic natural folds while using sequences that are distant from what is seen in nature [4–6]. These techniques take advantage of structural information and physical modeling, as well as statistical analysis of amino-acid conservation and co-evolution. Recent progress has also been made in the rational design of proteins with artificial folds from scratch [7–9]. All these computational design approaches nonetheless remain for now limited in their success and in the types of protein they can model.

 Machine learning methods provide an alternative and potentially complementary approach capable of exploiting the information available in protein sequence and structure databases. Natural sequence variation provides a rich source of information about the structural and biophysical constraints on amino acid sequence in functional proteins, but the unlabelled nature of much of the available data provides a challenge for straightforward supervised learning methods. The framework of generative modelling shows promise for exploiting this information in an unsupervised manner. Generative models are machine learning methods which seek to model the distribution underlying the data, allowing for the generation of novel samples with similar properties to those on which the model was trained [10]. In recent years, deep neural network based generative models such as Variational Autoencoders [11, 12], Generative Adversarial Networks [13], and deep autoregressive models [14–16] trained on large datasets of images [13], audio [15], text [17, 18], and even small molecules [19], have been shown to be capable of generating novel, realistic samples. Generative models can also easily be adapted to include auxiliary information to guide the generative process, by modelling the distribution of the data conditioned on the auxiliary variables. Such conditional generation is of particular interest for protein design where it is frequently desirable to maintain a particular function while modifying a property such as stability or solubility.

While there have recently been several successes in applying deep learning techniques to modelling protein sequences in tasks including contact prediction [20], secondary structure prediction [21, 22], and prediction of the fitness effects of mutations [23], the possibility of applying generative modelling methods in the design of new sequences has only very recently begun to be explored [24–31], and experimental evidence for the viability of these techniques is scarce. To realise the promise of generative models in protein engineering, work remains to be done in understanding the consequences of various design choices, the strengths and limitations of different types of model and the possibilities for integration into existing engineering workflows.

One particularly important consideration is the nature of the input representation to the model. Many traditional successes in protein sequence analysis have relied on features derived from multiple sequence alignments of related proteins, which simplify the inference of structural and functional constraints from sequence data [32]. Indeed, evolutionary information from alignments has previously proven useful in protein design and engineering [33–35]. However, alignments become large and unreliable as more distant proteins are added [36], placing an effective limit on the diversity of sequences that can be related in this way. For this reason, several recent works have explored deep learning methods which are capable of fully exploiting the data in sequence databases by working with raw sequence inputs. Deep sequence models such as LSTMs and transformers trained on datasets spanning the entire range of known sequences have been shown to learn representations which distill structural and functional information from the sequence [37, 38]. Despite these promising results, it remains unclear whether the representations learned are more informative than simple features computed from local alignments [39] and the generative capacity of these models, though acknowledged, is almost entirely unexplored.

Here, as a practical illustration of the application of deep generative design to protein engineering, we developed variational autoencoder (VAE) models capable of generating novel variants of bacterial luciferase, an enzyme which emits light through the oxidation of flavin mononucleotide (FMNH2). We proposed separate architectures to work with raw and aligned sequence input which, when trained on a family of almost 70000 luciferase-like protein sequences, learned representations capturing functional information at a variety of scales and generated novel sequences displaying patterns of amino acid usage characteristic of the family. Moreover, conditional versions of the models trained with auxiliary solubility information enabled control of the predicted solubility level of generated sequence variants. In order to confirm the generative capacity of the models, they were used to generate variants of the *luxA* subunit of the luciferase from *Photorhabdus luminescens*. A number of the variants generated by each model were selected for synthesis and assessed for function when expressed as recombinant proteins in *E. coli*.

## Results

### Generative VAE models for protein families

Variational autoencoders (VAEs) are deep latent variable models consisting of two subnetworks in an autoencoder structure [11]. The encoder network learns to map data points to low-dimensional 'latent vectors', while the decoder network learns to reconstruct data points from their low-dimensional latent encodings. Either raw or aligned protein sequences can be passed as input to a VAE model by representing them as fixed-size matrices whose columns contain 'one-hot encoded' representations of the identity of the amino acid at each position in the sequence (Fig 1). When trained on a training set of sequence inputs of the same kind, a VAE thus learns a latent representation of the content of each sequence. A prior enforcing

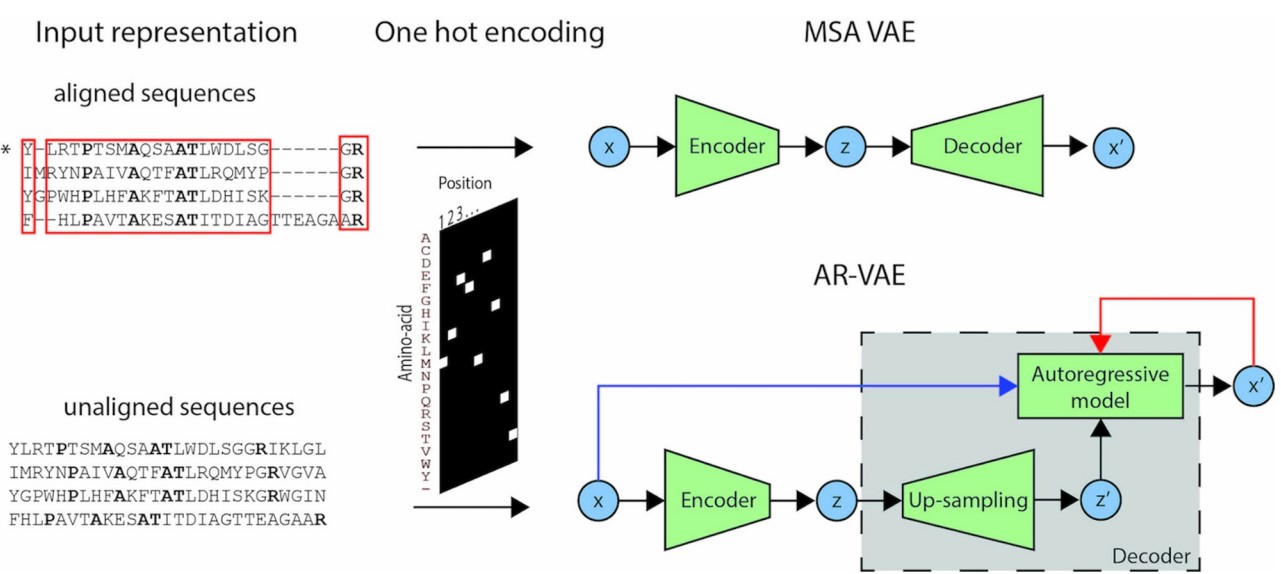

**Fig 1. Schematic representation of the input representation and VAE models used in the study.** Models take as input either raw or aligned sequences. In the latter case, the inputs correspond to the rows of an MSA of the luciferase family. Only columns of the MSA corresponding to positions (highlighted in red) present in the target protein (marked with *) are retained. In both cases, the sequences are one-hot encoded before being fed into the model. Different architectures were used depending on the type of sequence input. The model developed to work with aligned sequences (MSA VAE) used fully-connected feed-forward networks in both the encoder and the decoder. The model developed to work with raw sequences (AR-VAE) comprised a CNN encoder and a decoder which combined upsampling with autoregression. The decoder sequentially outputs predictions for the identity of the amino acid at each point in the sequence, conditioned on the upsampled latent representation together with the previous amino acids in either the input sequence (during training, blue arrow) or the generated sequence (when being used generatively, red arrow).

smoothness on the representations output by the encoder ensures that novel sequences can be generated either by varying the latent representation around that of existing sequences, or by sampling from the prior distribution over the latent vectors, and then feeding the resulting vectors through the decoder. This latent variable-governed generative process is particularly attractive for design applications because it can straightforwardly be used to bias generation towards particular regions of sequence space, either by sampling from the vicinity of the latent representations of target sequences, or by facilitating optimization based strategies which search the latent space for novel sequences with desirable properties [19, 40].

As a practical testbed for deep generative protein design, we chose to work with a dataset of 69,130 homologues of bacterial luciferase obtained from InterPro [41] (IPR011251). We worked with two versions of the dataset: one containing raw unaligned sequences, and one constructed from a multiple sequence alignment (MSA) of the dataset built using Clustal Omega (Materials and methods). Alignments of large protein families can be very wide, presenting a challenge for methods seeking to model variation at all positions. We chose instead to build models capable of generating variants of a single target protein, the luciferase *luxA* subunit from *P. luminescens*. We therefore dropped all columns of the MSA which were unoccupied in the *luxA* target. We split the dataset into a training set and a holdout validation set, using the same split for both aligned and raw sequences. In order to avoid highly similar sequences occurring in the training and validation sets, we first clustered all the sequences using mmseqs2 [42], and then added clusters chosen at random to the validation set until the total number of sequences in the validation clusters reached 20% of the total. In order to assess generalisation to a range of distances from the training set, three train-validation splits were created using sequence identity thresholds of 30%, 50% and 70% in the clustering. Since our ultimate goal was the generation of variants with reasonably close similarity to the target

protein, we mainly used the split at a clustering threshold of 70% sequence identity for the development of models, but report amino acid reconstruction accuracies on all three splits in S1 Table.

Following models previously developed to model the fitness consequences of mutations [23, 43], we used a standard design of fully connected feed-forward encoder and decoder networks for the models taking aligned input (MSA VAE, Materials and methods). Preliminary experiments with a similar architecture on unaligned sequence data yielded poor results, with the generated sequences often failing to register as hits when scored with the family profile HMM from PFAM [44]. In a VAE with a feed-forward decoder, the output variables are conditionally independent given the latent variables, meaning that all information about local conditional dependencies must be stored implicitly in the latent variables. The importance of capturing such local dependencies in unaligned sequence data makes autoregressive models such as recurrent neural networks (RNNs), which can be trained to explicitly model the relevant conditional distributions, a natural choice. VAEs can be enhanced with autoregressive decoders to reduce the burden on the latent space to capture local information, and architectures based on this principle have been used to model images, text and molecules [16, 17, 19, 45].

To handle raw sequences we therefore designed a model incorporating a convolutional encoder as well as a hybrid decoder [46] containing feed-forward and autoregressive components (AR-VAE, Materials and methods). We found that this hybrid structure was crucial in allowing the model to fit sequences containing hundreds of amino acids, and helped ensure that the latent space was used, partially circumventing the well-documented optimization difficulties that arise when training VAE models with autoregressive decoders [17, 46]. As an initial confirmation of the advantages of the chosen architecture, we scored a set of 3000 sequences generated by sampling from the prior of the AR-VAE model with the family's profile HMM. As baselines we also computed HMM scores for sets of sequences generated by the MSA VAE model, and by a model having the same architecture as MSA VAE trained on raw sequence data. The vast majority of sequences generated by both MSA-VAE and AR-VAE were scored as hits by the HMM (96.8% and 99.7% respectively, at an E-value threshold of 0.001), whereas the sequences generated by the baseline model trained on raw inputs only scored as hits just over half the time (57.3%).

## Models learn representations encoding features relevant to biological function

To model the distribution of sequences within a protein family, VAEs develop internal representations of the content of sequences at multiple resolutions. To explore the biological significance of these representations we first examined the weights in the output layer of the decoder. At each point in the sequence this layer is parameterised by a weight matrix whose columns represent learned 'embeddings' of amino acid identity, which combine with the network's hidden representation via a softmax transformation to output the probabilities of observing each amino acid at that point. If the weights of this layer are tied across all positions, as was the case for AR-VAE models, a single set of embeddings is obtained. Visual inspection of a two dimensional projection of these embeddings obtained using PCA indicates that they reflect the biochemical properties of the various amino acids: for example the negatively charged amino acids (D and E) and the positively charged amino acids (K, R and H) cluster tightly together (Fig 2), while the second principal component seems to separate the hydrophobic amino acids (both the hydrophobic and aromatic groups in the legend) from the polar amino acids, recapitulating the major groupings in traditional classification schemes [47]. As further validation of

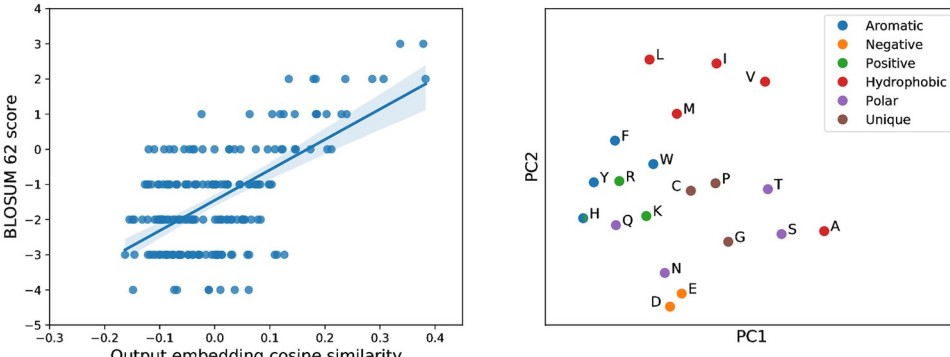

**Fig 2. Amino acid representations learnt by VAE models capture biochemical properties.** Left: pairwise cosine similarities between amino-acid output embeddings from an AR-VAE model trained on unaligned sequences correlate with amino acid substitution scores in the BLOSUM 62 substitution matrix (Spearman $\rho$ = 0.423, $n$ = 190); right: projection of AR-VAE output embedding weights onto first two principal components groups embeddings corresponding to biochemically related amino acids.

the biological relevance of these embeddings we found that the cosine similarities between embeddings for pairs of amino acids were correlated with the entries in the BLOSUM 62 substitution matrix (Fig 2). Finally, to understand the models' representations at a more global level, we examined the distribution of latent vectors associated with sequences coming from distinct sub-families within the set of luciferase-like proteins. The InterPro sub-families form visually distinct clusters in the space of the first two principal components, especially for the model trained on the MSA (Fig 3), indicating that global information about functional and evolutionary relationships between sequences is captured in the latent variables.

## Models reproduce patterns of amino acid usage characteristic of members of the family

Protein families are characterised by statistical features that reflect the shared evolutionary history and related structure and function of members of the family. Patterns of amino acid conservation at individual positions reflect the presence of functionally important sites and are used by profile HMMs to identify family members [48], while correlations in amino acid usage

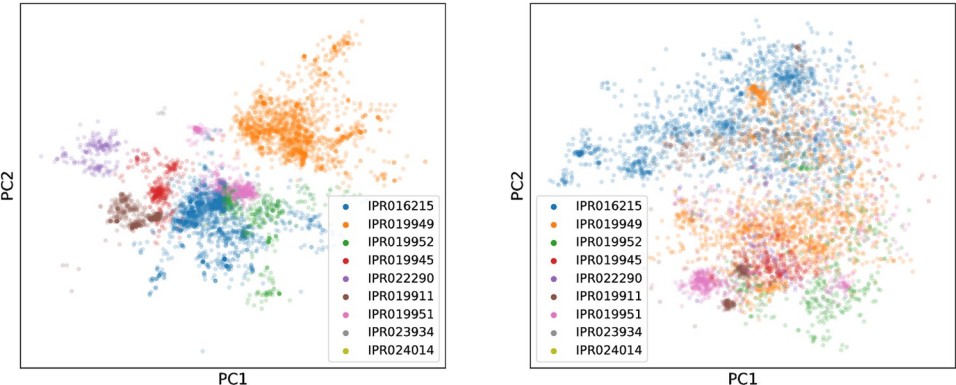

**Fig 3. Organization of latent space reflects functional groupings.** Visualisation of the latent representation of validation set sequences for MSA VAE (left) and AR-VAE (right), projected onto first two prinicipal components and coloured by sub-family annotation derived from InterPro. Only sequences belonging to one of the 9 largest sub-families are shown.

at pairs of positions are signatures of structurally constrained evolutionary covariation which can be used to infer contacts between residues [49–51]. Models such as VAEs which seek to learn the distribution of sequences in the family can be evaluated for their capacity to reproduce these characteristic statistical features. To further probe the ability of the AR-VAE and MSA VAE models to generate realistic sequences, we therefore calculated first- and second-order amino acid statistics from the sets of 3000 sequences previously generated by sampling from the prior of each model and compared them to corresponding statistics calculated from the sequences in the training set. Making a comparison of these statistics requires an alignment of the generated sequences to the training sequences. Such an alignment is automatically available for MSA VAE; for AR-VAE we used Clustal Omega to jointly align all training and generated sequences, and again filtered columns based on the alignment of the target *luxA* sequence. Given these alignments, we computed single-site amino acid frequencies at all positions and pairwise amino acid frequencies and covariances [32, 52] at all pairs of positions for both the subset of the alignment corresponding to generated sequences and the subset corresponding to training sequences (Fig 4). Both VAE models were able to reproduce the statistics observed in the natural sequences reasonably well, with the MSA VAE sequences showing especially good agreement. As a simple baseline, we also sampled a set of 3000 sequences from the PFAM profile HMM for the family, and compared statistics at match states to statistics at positions assigned to match states when aligning the training set to the model. By construction, HMM models ignore interactions between residues, and therefore generate sequences which show similarity to natural sequences in first order statistics (patterns of conservation) but whose covariances are (approximately) zero. We note that detailed direct comparison of the results between models is challenging due to the statistics being computed from different model-specific alignments, and in the case of the HMM model, due to the fact that it was not trained on the same data. Nonetheless, the analysis serves to illustrates the fact that the VAE models, in contrast to simpler profile models, are able to reproduce second-order statistics without being fit to them directly, and, in the case of AR-VAE, without requiring aligned input data.

To obtain a more qualitative understanding of the kinds of dependencies that the models were able to capture, we used the direct coupling analysis software CCMPred [51] to identify the most strongly 'coupled' pairs of positions in the generated sequences. Direct coupling analysis seeks to explain the observed (first and second-order) amino acid statistics in terms of couplings between positions in a statistical model [32]. The most strongly coupled pairs of positions in natural family alignments are good predictors of contacts in protein 3D structure [50, 51]. We therefore compared the couplings inferred from generated sequences to the contacts in the 3D structure of *luxA* by visualising the resulting predicted contact maps (Fig 5). Whereas the MSA VAE generated sequences which exhibited strong dependencies between positions at a range of distances, yielding an inferred contact map bearing a reasonable resemblance to the ground truth, the sequences generated by the AR-VAE showed a bias towards local couplings.

## Experimental validation of generated sequences

**Novel luminescent variants generated from latent vicinity of a target luminescent protein.**　　Bacterial luciferase is a heterodimeric enzyme which catalyzes the light-emitting reaction responsible for the bioluminescence of a number of bacterial species. The two homologous subunits, encoded by the genes *luxA* and *luxB*, have different roles: the *luxA* subunit contains the active site, while the *luxB* subunit is thought to provide conformational stability [53]. Since the luciferase activity is primarily encoded by the *luxA* gene, we sought to generate

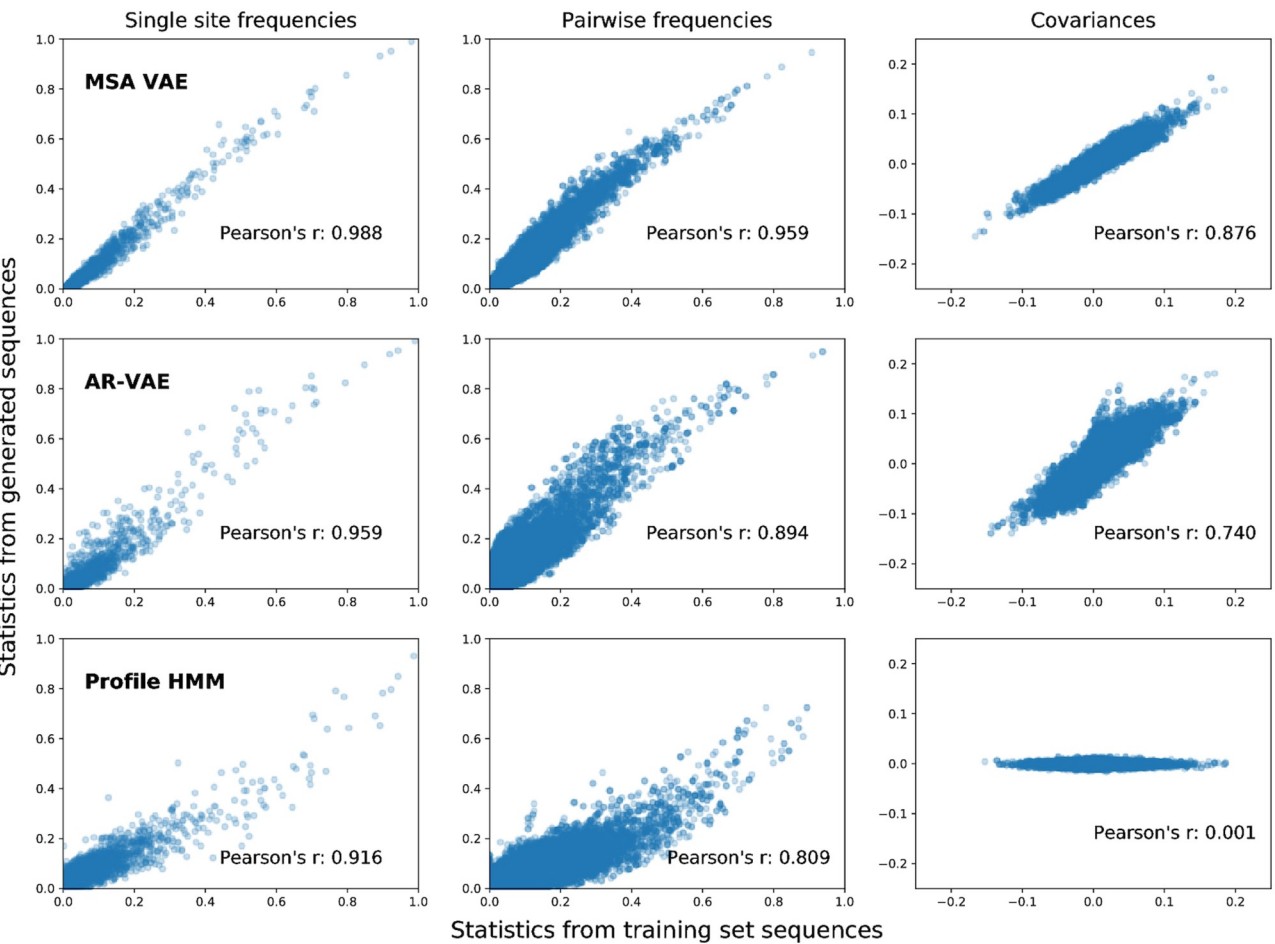

**Fig 4. Statistics computed from alignments of generated sequences to natural sequences from the training set.** Similarity of statistics between generated and natural sequences reflect the ability of models to capture important types of sequence variation. Single-site amino acid frequencies (left) capture patterns of residue conservation at each position in the alignment, while co-occurrence frequencies (centre) and covariances (right) between amino acid identities at different pairs of positions reflect patterns of evolutionary covariation which may indicate structural or functional constraints. Sequences were generated by sampling from the prior of the VAE models. For MSA VAE the resulting sequences were already aligned; for the raw sequences generated by AR-VAE, a new MSA was first constructed by running Clustal Omega on the set of sequences sampled from the model together with the natural sequences in the training set, using the bacterial luciferase family PFAM profile HMM as an External Profile Alignment, following which statistics for generated and natural sequences were computed from the corresponding subsets of the alignment. As a baseline we also report results for statistics generated by the profile HMM from PFAM. In this case the training set statistics were computed from the alignment of the training sequences to the profile HMM.

novel variants of the *luxA* subunit, taking as our seed sequence the *luxA* protein from the species *P. luminescens* (UniProt id: P19839). For both AR-VAE and MSA VAE models we generated a set of candidate variants by sampling latent vectors from the neighbourhood of the latent space encoding of P19839 (Materials and methods) and passing them through the decoder. To validate these candidates, 12 sequences from each model were selected for synthesis (S1 File), spanning a range of distances (17-48 total differences including substitutions and deletions) to P19839.

To assess the activity of the generated variants, the sequences were synthesised and expressed from a plasmid in an *E. coli* strain carrying the *luxCDBE* genes on a second plasmid. 9 of the 11 successfully synthesised sequences generated by the MSA VAE showed measurable luminescent activity, compared to 6 of 12 generated by the model trained on unaligned sequences (Fig 6). Furthermore, the MSA VAE sequences showed a level of luminescence

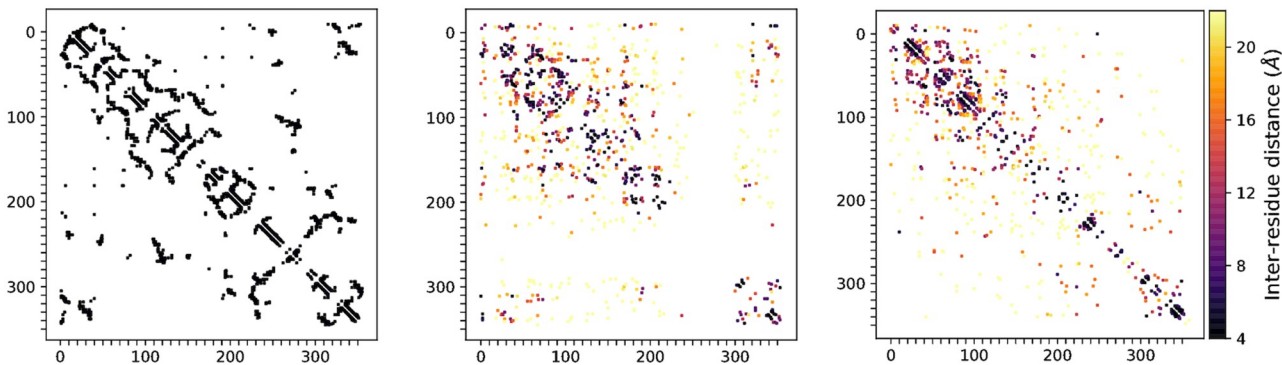

**Fig 5. Comparison of inter-residue couplings inferred from generated sequences to contacts in the 3D structure of a *luxA* protein.** Left: contact map of *luxA*, showing 1000 closest contacts separated by at least 4 sequence positions. Centre and right: top 1000 couplings inferred from sequences generated by MSA VAE and AR-VAE respectively, coloured by distance between residues in *luxA* 3D structure. Couplings were predicted using CCMPred on samples of 3000 sequences, and only couplings between residues separated by at least 4 sequence positions were shown. The patterns of inferred couplings reflect the dependencies captured by the models: while MSA VAE captures realistic dependencies between positions at a range of distances, the sequences generated by AR-VAE exhibit a bias towards local dependencies.

comparable to that of the wild-type protein, while the AR-VAE sequences tended to have reduced luminescence. Remarkably, there was no evidence in a drop-off in luminescence as the distance from the wild type increased for the sequences generated by the model trained on the MSA. This was not true for sequences generated by the model trained on the unaligned family members. Comparison of the generated sequences to other *luxA* sequences from the training set revealed that several of the more distant variants from both models were closer to other training set sequences than they were to the seed *P. luminescens luxA*. This was especially true for the MSA VAE, indicating that this model's latent space is organised in such a way as to encourage the exploration of functional regions of sequence space lying between existing

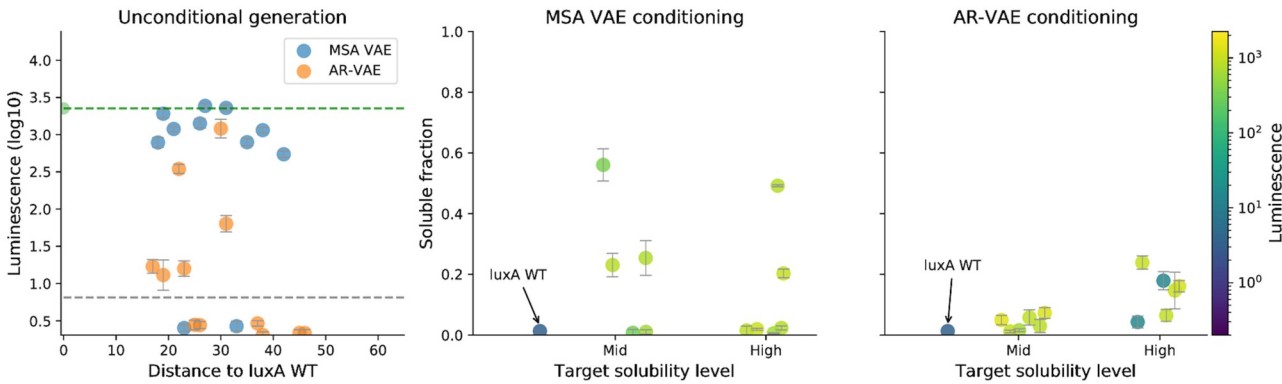

**Fig 6. Luminescence measurements for synthesised protein sequences generated from latent vectors sampled from the neighbourhood of the encoding of the *P. luminescens luxA* sequence.** Left: luminescence of sequences generated by VAE models trained on raw (AR-VAE) or aligned (MSA VAE) sequences from the family of luciferase-like proteins (mean across fifteen replicates, error bars represent standard deviation). Wild-type sequence luminescence is displayed as a dashed green line. The dashed grey line represents the detection threshold, conservatively set to twice the mean untransformed luminescence of a strain lacking *luxA*. Distance is computed as number of substitutions and indels relative to wild type. The MSA VAE model was able to generate functional sequences with large numbers of differences to wild type, whereas the AR-VAE model seemed to introduce deleterious mutations more rapidly. Center and right: measurements of both solubility and luminescence for sequences generated by VAE models conditioned on predicted solubility level show that conditional models can be used to engineer increased-solubility variants of a *luxA* sequence while preserving function. Solubility is reported as the ratio of the amount of protein present in the supernatant to the total amount in both supernatant and pellet of lysed *E. coli* cells over-expressing the protein, as measured by a dot blot assay (mean of four technical replicates, error bars represent standard deviation).

sequences. Even taking this into account, the luminescent MSA VAE variants were all between 18-35 substitutions (including deletions) from any training set sequence.

**Conditional VAEs enable enhancement of solubility of a *luxA* protein.** In order to assess the ability of our models to generate novel functional sequences with specified biophysical properties we further sought to use conditional variants of the VAE models to increase the solubility of the P19839 *luxA* sequence. Proteins frequently aggregate and precipitate when expressed at high concentrations [54]. This phenomenon is a challenge in a wide range of applications, from the production of protein therapeutics to the study of protein biochemistry and structure, leading to interest in engineering of increased-solubility variants [55]. We considered P19839 to be a suitable target to test the use of conditional VAE models for solubility engineering as it was predicted to be insoluble by a recent sequence-based computational solubility prediction method, protein-sol [56], with subsequent experimentation confirming that it was indeed recovered in the insoluble fraction when expressed in *E. coli* (S1 Fig).

Training sequences were grouped into three equally-sized bins by predicted solubility value calculated using protein-sol and the bin label was used as the conditioning variable when training conditional versions of both AR-VAE and MSA VAE models, corresponding to a specification of either low, medium or high solubility for the sequence. P19839 was assigned to the low solubility bin. In order to generate variants with increased solubility, we sampled latent vectors from the neighbourhood of the encoding of P19839 and passed them through the decoder together with the conditioning variable, which was fixed to a value corresponding to either medium or high solubility. To check that conditioning was successful, we generated 100 sequences at each solubility level from the conditional AR-VAE and MSA VAE models, and calculated predicted solubility values for the new sequences (Fig 7). Both models were able to control the predicted solubility level fairly successfully while introducing only relatively few additional mutations compared to the original decoding. Moreover, comparison with the predicted solubility of P19839's closest neighbours in the training dataset reveals that the conditioned sequences are predicted to be much more soluble than any training dataset sequence at an equivalent distance to P19839 (S2 Fig). Repeating similar analyses for models trained with different random seeds, we found that while there was some variability in the level of predicted solubility of variants across seeds for both conditional and unconditional models, conditional models consistently enabled sampling of higher solubility variants (S3 Fig).

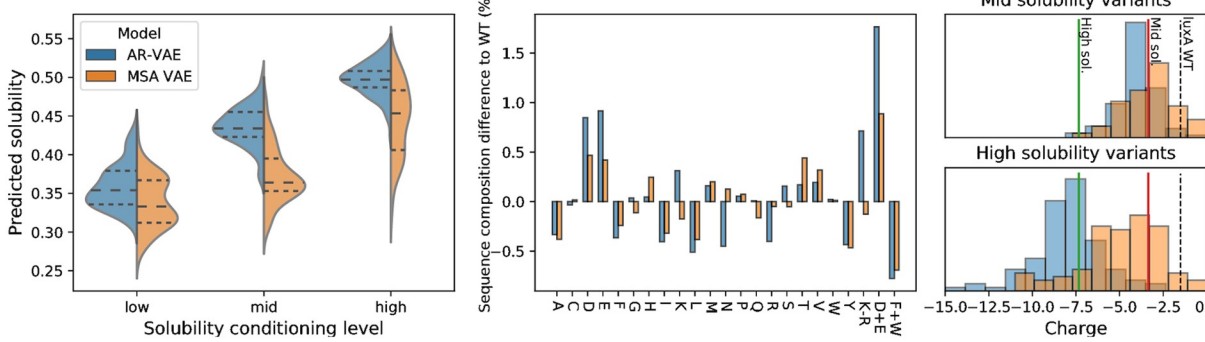

**Fig 7. Computational analysis of variants generated by conditional VAE models conditioned on predicted solubility level.** Left: distribution of predicted solubilities of sequences generated when conditioning on each of three solubility levels (median and upper and lower quartiles indicated with horizontal lines); centre: difference in amino acid composition percentages between generated variants at highest solubility level and original P19839 *luxA* sequence, including values for combined amino acid features used in protein-sol prediction algorithm; right: distribution of charge of *luxA* variants generated by conditioning on high (top) and medium (bottom) solubility levels. For comparison, the charge of the original P19839 *luxA* sequence is shown as a dashed line, the average charge for high solubility sequences in the training set is shown as a solid green line, and the average charge for medium solubility sequences in the training set is shown as a solid red line).

To understand the changes being made by the conditional VAE models to increase protein-sol's predicted solubility values, we computed several of the features used as inputs by protein-sol for both the generated sequences and training sequences. The features for generated sequences tended to have values which were shifted from P19839's values towards those exhibited on average by soluble sequences from the training set (Fig 7). For example, when asked to produce sequences at the highest solubility level, the models produced sequences with more negative charge than P19839, by favouring substitutions of neutral or positively charged residues with negatively charged aspartic (D) and glutamic (E) acids (Fig 7).

Finally, we randomly selected 6 sequences from each of the two increased solubility levels for each model for synthesis, and measured luminescence as well as solubility. To measure solubility, the protein variants were cloned on a pET28/16 plasmid with a His tag and expressed in Rosetta E. coli cells for 3H, followed by mechanical lysis and centrifugation. The amount of *luxA* protein present in the supernatant and in the pellet were measured by performing a dot blot assay using an anti-HisTag antibody, and the average fraction in the supernatant across four replicates was taken as a measure of solubility. 5 out of 12 sequences generated by the MSA VAE model showed clearly improved solubility and 4 out of these maintained a high luminescence level (Fig 6). Almost all sequences generated by AR-VAE showed signs of improved solubility, and out of the 4 sequences with a considerable fraction ($>10\%$) in the supernatant, 3 maintained a high luminescence level. In both cases these improvements were achieved while introducing only a relatively small number of mutations to the wild type (12-26), highlighting that conditioning separated information about solubility from information about general protein content successfully enough to be useful for fairly sensitive engineering.

**Model likelihoods predict observed luminescence values.** A crucial assumption in the use of generative models for design is that by learning the distribution of sequences in a particular family, the model captures patterns of sequence variation that underlie function. In practice, this is achieved by training the model in such a way as to maximise the likelihood of the sequences in the family, since these were arrived at by a process of natural selection operating on a common or related set of functional constraints. If quantitative measurements of function are available for a (heldout) set of mutated family members, then the relationship between the likelihood assigned to these sequences by the trained model and their observed functional values can be used to test this assumption [23, 29, 57, 58]. Previous work adopting this approach for the unsupervised prediction of the fitness effects of mutations has demonstrated state of the art performance using VAE models trained on MSAs [23].

As a final test of the ability of our models to capture functional constraints, we therefore sought to use the full set of synthesised sequences and corresponding luminescence measurements to evaluate the extent to which the likelihoods of our models were predictive of the experimentally determined luminescence values. To score the sequences we retrained MSA VAE and AR-VAE models with three different random seeds, to avoid bias when comparing sequences generated by different methods due to allowing a model to score its own generations. For both the MSA VAE and AR-VAE models we computed approximations to the likelihood for each sequence via the ELBO (Materials and methods). As baselines we additionally considered scores obtained from the BLOSUM 62 substitution matrix, and the PFAM profile HMM for the family (Materials and methods). The scores from the VAE models and baselines were compared to the experimentally determined luminescence values for the full set of sequences generated by both unconditional and conditional models, together with the *luxA* wild-type sequence.

To assess the models' scores we computed three metrics: two measures of the rank correlation between the scores and the observed luminescence values, and one classification metric. For the former, as well as the Spearman rank correlation we report a nonlinear rank

**Table 1. Metrics assessing ability of model scores to predict experimental luminescence values.**

|  | AUC | Spearman | Mapped |
|---|---|---|---|
| BLOSUM score | 0.57 | 0.14 | 0.28 |
| PFAM HMM score | 0.76 | 0.50 | 0.45 |
| AR-VAE ELBO | 0.92 | 0.57 | 0.76 |
| MSA VAE ELBO | **0.96** | **0.61** | **0.79** |

Scores derived from VAE and baseline models were compared to experimentally determined luminescence measurements for all tested variants. The AUC metric measures the ability of the model scores to distinguish between luminescent and non-luminescent sequences, where here all sequences with less than 5% of the raw WT luminescence level are included in the non-luminescent group, to approximately reflect the bimodality in the measured values. 'Mapped' refers to the robust rank correlation described in [57], which maps model scores to the experimental scale by assigning the sequence with the $n^{th}$ highest score the $n^{th}$ highest experimental value, then computes the Pearson correlation. This metric is more sensitive to bimodal data that the Spearman rank correlation. When computing it, we use the $log_{10}$-transformed luminescence values. For the VAE models, the reported metric is the mean across identical models trained with three different random seeds.

correlation that is more sensitive to the bimodality of the data, and whose use has previously been suggested in this context [57]. For the latter we report an AUC score measuring the ability of the models to separate the sequences into luminescent and non-luminescent. The results are shown in Table 1. Across all metrics, the likelihoods of both VAE models perform better than either of the baseline scores, with the MSA VAE likelihoods the highest scoring in each case, demonstrating the ability of this model to learn functional constraints from natural sequence variation.

## Discussion

We have developed variational autoencoder models capable of generating novel functional variants of a luminescent protein when trained on a set of homologues of the target. Computational analysis of separate VAE models developed for raw sequences and aligned sequences suggested that the version trained on MSA data more plausibly reproduced the statistical features characteristic of the structural and functional constraints on members of the family arrived at and maintained over the course of evolution. Experimental validation confirmed that a significant fraction of variants of a target *luxA* protein generated by both models were functional, while confirming the strengths of the MSA model, which generated a set of variants which almost without exception retained high levels of luminescence despite diverging by as many as 35 amino acid differences from any protein in the dataset.

The application of generative models to multiple alignments of protein families is not new. Markov random field models with pairwise couplings between residues are the basis of the most successful unsupervised contact prediction methods [50, 51]. Recent advances in efficient inference methods for these models have been shown to permit the generation of sequences which accurately reproduce the low-order statistics from natural MSAs [52, 59], and a concurrent work has demonstrated the successful application of these methods to protein design [60]. However, such models are restricted to modelling low-order dependencies among variables and require aligned inputs, motivating the exploration of more flexible and expressive classes of model. VAEs are able to capture higher-order relationships between variables, can be modified to straightforwardly and flexibly incorporate conditioning information, and learn latent spaces which offer various possibilities for controlled generation, including the local sampling strategy employed here. In cases where additional data relating sequences to some property of

interest is available, the continuous latent space facilitates the conversion of (discrete) sequence optimisation problems to more straightforward continuous optimisation problems [19]. Finally, VAEs can be adapted to handle raw sequence data, permitting the generation of full-length sequences and offering the possibility of training on sequence data from multiple families.

Previously, it was shown that VAE models trained on multiple sequence alignments of single protein families could be used to predict the fitness consequences of mutations, by comparing the approximate likelihoods of mutant and wild-type sequences under the model [23, 43]. Unlike these prior works we focus on the generative capabilities of VAE models, and, when working with aligned sequence input, filter the alignment in such a way as to allow the generation of full variant sequences of a single target protein. While we found VAEs trained on aligned sequence data to be effective at reliably generating functional variants at a range of distances to a target protein, there are nonetheless shortcomings to this approach. Building a training dataset for this model requires the construction of a large multiple sequence alignment. Even where sufficient related sequences are available this poses challenges. Such alignments will often have a very large number of columns, and while a relevant subset of columns can be retained, as done here, this restricts the sequence variety that can be generated, since only one or a handful of sequences will be fully represented in the retained columns. Moreover, the construction of large alignments remains a difficult problem with a trade-off in this context between the decline in alignment accuracy associated with the arbitrary addition of extra homologous sequences [36] and the desire to use large numbers of sequences to exploit the capacity of the model and fit it reliably.

Here we chose to work with a relatively large family, and sought to exploit this diversity to fit expressive models. Nonetheless, retraining identical models on randomly selected subsets of the training data, we found that models trained on only thousands of sequences still successfully distinguished between luminescent and non-luminescent sequences (S4 Fig), indicating that such techniques are applicable beyond the large family studied here. Indeed, prior work fitting VAEs on MSA data has demonstrated strong results across a range of family sizes, albeit with the additional use of biologically motivated priors and Bayesian treatment of the model weights [23].

The possibility of sidestepping the issues associated with alignments is a key advantage of models trained on raw sequence data. In principle such models would make it possible to leverage the full variety of protein sequence information by, for example, pretraining models on entire protein sequence databases [30, 37, 38, 61]; fine-tuning of such models on individual families has shown promise for some prediction tasks and may facilitate generalisation to smaller sequence families [30, 37, 38, 62], though application of similar techniques to generative models for sequence design remains to be demonstrated. Conditional VAEs with feed-forward decoders have previously been used to generate sequences with specified metal binding sites or structural topologies when trained on raw sequences spanning multiple families [26]. Other recent work has used deep autoregressive models without latent variables to handle raw sequence data, inspired by similar approaches in natural language processing [29, 30, 37]. Here, seeking to combine the advantages of latent variable models and autoregressive models, we showed that a VAE with an autoregressive decoder could be used to generate realistic sequences when trained on the members of a single family, and provided experimental validation of the function of a number of generated *luxA* variants. However, we also found that this model seemed to be less effective at capturing the long-range dependencies between amino acids at different positions than the MSA-based model. Closing this performance gap is an important challenge for future work if the potential advantages of training models on raw sequence data are to be fully realised. Other sequence models such as transformers might be

better suited to capturing long-range interactions and have already shown promise in modelling proteins [38, 39, 61]. More fundamentally, evaluating the capability of alternative kinds of model *in silico* requires quantitative measures of the quality of generated sequences, and this, too, remains a difficult problem.

The successful generation of full, functional variants at a range of distances to a given engineering target opens the door to a multitude of applications in the field of protein design. Here we showed that using conditional variants of the models it was possible to generate new variants with modified biophysical properties in a controlled way, by generating variants of a *luxA* protein with increased solubility relative to wild-type. More generally, a model capable of guiding exploration of distant regions of functional sequence space could be used to significantly improve the efficiency of existing design approaches [63], or to constrain the search of sequence space for proteins with desirable properties [19, 40]. The ability to generate novel sequences with a desired function is an important desideratum in protein engineering approaches, and while here we have shown that straightforward conditioned generation is sufficient to generate novel, functional sequences satisfying basic biochemical criteria, we expect that combining these kinds of methods with existing engineering techniques will result in even more powerful and widely applicable methods for protein sequence design.

## Materials and methods

### Dataset construction

**Selection of sequences.**    All sequences containing a luciferase-like domain (IPR011251) were downloaded from InterPro [41]. Sequences longer than 504 amino acids were discarded. The remaining 69130 sequences were clustered using mmseqs2 [42] with a sequence identity threshold of 70%. To create a validation set, clusters were randomly removed from the training set until the number of sequences in all of the removed clusters was 20% of the total. The same train/validation split was used for models irrespective of whether they took as input aligned or unaligned versions of the sequences.

**Multiple sequence alignment.**    To create a multiple sequence alignment from the dataset, the full set of training and validation sequences were aligned using Clustal Omega [36] using the profile HMM of the bacterial luciferase family from PFAM [44] as an external profile alignment to guide the creation of the MSA. The resulting MSA was very wide, presenting potential modelling challenges. To circumvent these, only a subset of columns were retained on the basis of the target protein (details below).

**Input representation.**    All sequences are represented as fixed size matrices by one-hot encoding the amino acids, such that a sequence of length $L$ is represented by a $L \times 21$ matrix (a gap/padding character is used together with the 20 standard amino acids). When raw sequences are used as input, a fixed input size is ensured by right padding sequences up to a length 504 (and dropping sequences exceeding this length). When aligned sequences are used as input, all columns in the MSA which are assigned gaps in the alignment of the target *luxA* protein P19839 are dropped, leaving 360 columns.

### Variational autoencoders

VAEs [11] posit a set of latent variables $\mathbf{z}$ associated with each input $\mathbf{x}$ and model the joint distribution $p(\mathbf{x}, \mathbf{z}) = p_\theta(\mathbf{x}|\mathbf{z})p(\mathbf{z})$ of the latents and the observations. The distribution $p_\theta(\mathbf{x}|\mathbf{z})$ over the values of the observed variables given the latents is parametrised by a neural network (the 'decoder') with weights $\theta$, and $p(\mathbf{z})$ is a prior over the latents, typically chosen to be a factorised Gaussian distribution. An inference model $q_\phi(\mathbf{z}|\mathbf{x})$ parametrised by a second neural network

(the 'encoder') is introduced to approximate the intractable posterior $p_\theta(\mathbf{z}|\mathbf{x}) = \frac{p_\theta(\mathbf{x}|\mathbf{z})p(\mathbf{z})}{\int p_\theta(\mathbf{x}|\mathbf{z})p(\mathbf{z})d\mathbf{z}}$,

allowing the construction of a training objective representing a lower bound on the log-likelihood (the Evidence Lower Bound or ELBO):

$$\mathcal{L}(\phi, \theta; \mathbf{x}) = \mathbb{E}_{q_\phi(\mathbf{z}|\mathbf{x})}[\log p_\theta(\mathbf{x}|\mathbf{z}) - D_{KL}(q_\phi(\mathbf{z}|\mathbf{x})||p(\mathbf{z}))] \leq \log p_\theta(\mathbf{x}) \ . \tag{1}$$

Jointly maximising this objective over a set of training examples with respect to the weights of the two networks enables the generative model and the inference model to be learned simultaneously.

The VAE framework offers flexibility in the architectures of the encoder and decoder networks. In a standard setup in which feed-forward networks are used for both encoder and decoder, the observed variables are conditionally independent given the latents. A more flexible output distribution can be obtained by instead decoding autoregressively [17, 45]. That is, given a latent vector $\mathbf{z}$, the output sequence $\mathbf{x} = (\mathbf{x_1}, \ldots, \mathbf{x_L})$ is generated one position at a time, with the decoder modelling the conditional distributions $p_\theta(\mathbf{x_i}|\mathbf{x_1}, \ldots, \mathbf{x_{i-1}}, \mathbf{z})$ of each output $\mathbf{x_i}$ given the values of its predecessors and the latents. This corresponds to modelling the distribution of $\mathbf{x}$ given $\mathbf{z}$ in the factorised form $p_\theta(\mathbf{x}|\mathbf{z}) = \prod_i p_\theta(\mathbf{x_i}|\mathbf{x_1}, \ldots, \mathbf{x_{i-1}}, \mathbf{z})$.

A VAE can straightforwardly be adapted to model the distribution of the data conditioned on auxiliary variables $\mathbf{c}$ by conditioning the encoder and decoder networks on these variables [64]. The objective then becomes

$$\mathcal{L}(\phi, \theta; \mathbf{x}) = \mathbb{E}_{q_\phi(\mathbf{z}|\mathbf{x},\mathbf{c})}[\log p_\theta(\mathbf{x}|\mathbf{z}, \mathbf{c}) - D_{KL}(q_\phi(\mathbf{z}|\mathbf{x}, \mathbf{c})||p(\mathbf{z}|\mathbf{c}))] \ . \tag{2}$$

## Model architecture

**MSA VAE.** Both the encoder and the decoder are fully connected neural networks with two hidden layers. We experimented with a range of layer sizes and latent dimensions, settling on 256 units per hidden layer and a latent dimensionality of 10 unless otherwise specified. ReLU activations were used for hidden units, and softmax activations for the output units.

**AR-VAE.** We used a convolutional neural network for the encoder, consisting of 5 layers of 1D convolutions of width 2. Apart from the first layer, the convolutions were applied with a stride of 2. The first layer used 21 filters; this number was doubled in each successive layer. PReLU activations were used, and batch normalization was applied in each layer. The size of the latent dimension was 50.

The decoder consisted of two components, similar to the decoder in a 'hybrid' autoregressive VAE model developed for text [46]: an 'upsampling' component, which contained 3 layers of transposed convolutions to 'upsample' the latent vector to a sequence of the same length as the output sequence; and an autoregressive component, which was a GRU with 512 units which took as input at each timestep the full sequence of previous amino-acids and upsampled latent information, and output the predicted identity of the amino acid at the next timestep.

Optimization difficulties have been reported when training VAEs with powerful autoregressive decoders [17, 46]. To address these we followed [17] in applying dropout to the amino acid context supplied as input to the GRU during training, such that 45% of the amino acid context was masked out, forcing the network to rely on the information transmitted via the upsampled latent code together with the conditional information in the masked amino acid sequence to make its prediction.

**Conditioning on predicted solubility.** Solubility predictions were made for all proteins by running the protein-sol [56] software on the full sequences. The resulting solubility predictions were continuous values ranging between 0 and 1. We discretized this information by

binning the sequences into 3 equally-sized bins corresponding to low, mid and high solubility. Bin membership was fed as a one-hot encoded categorical variable as additional input to both encoder and decoder in conditional versions of the models.

### Details of model training and selection procedures

Models were trained using SGD with a batch size of 32 and the validation set was used to monitor performance at the end of each epoch as measured by ELBO loss and amino-acid reconstruction accuracy. Unless otherwise specified, the Adam optimizer was used with a learning rate of 0.001. We also monitored the reconstruction accuracy of P19839, the *luxA* protein which had been selected for synthesis. This single-datapoint reconstruction accuracy was considered when choosing model hyperparameters together with the other two metrics and the evaluations described above. In particular, weights from the epoch which showed the best reconstruction accuracy of P19839 without evidence of overfitting (i.e. increase in validation loss) were saved and used to generate the variants of P19839 that were tested experimentally.

### Computational analysis of generated sequences

For each model, a set of 3000 sequences was generated by sampling latent vectors from the prior and decoding greedily. Before further analysis, we constructed an alignment of the sequences generated by AR-VAE to the training sequences, following the procedure used to create the training alignment for MSA VAE. In detail, we ran Clustal Omega on the sequences from the training set together with the generated sequences, using the profile HMM to guide the alignment, and subsequently dropped all columns unoccupied in the row corresponding to the P19839 *luxA* sequence. We used the EVCouplings python package [65] to compute both single site amino acid frequencies and pairwise co-occurrence frequencies separately for the aligned generated sequences and the aligned training sequences. Gap frequencies were not included in the comparison between generated and training statistics. To infer couplings we ran CCMPred on the aligned generated sequences. We used EVCouplings to compare the resulting couplings to contacts in the 3D structure of the *luxA* protein, aggregating structure information from three luxA structures in PDB (1luc, 3fgc and 1brl).

**Scoring of sequences with experimentally determined luminescence values.**   The likelihood of the VAE models was approximated via the ELBO, with 200 samples used to compute an average ELBO for each sequence. As baselines we took the score obtained by summing the BLOSUM scores of all substitutions, and the log-odds score for the sequence returned by scoring it with the PFAM profile HMM for the family of luciferase-like proteins. HMMER was used to compute log odds scores.

### Selection of variants for synthesis

In total 48 sequences were synthesised and tested for function as luciferase *luxA* subunits. These corresponded to 12 sequences for each of four models: unconditional and conditional versions of the MSA VAE trained on aligned sequences, and unconditional and conditional versions of the AR-VAE model trained on raw sequences.

**Unconditional generation.**   Unconditional models trained on aligned and unaligned sequence data were used to generate sets of candidate luminescent *luxA* proteins. 12 sets of sequences were selected for synthesis from each model. To generate the sequences, first the one-hot encoded P19839 *luxA* sequence was passed through the encoder of the VAE model to obtain the mean and variance for the factorised Gaussian posterior distribution over the latent variables. To enhance diversity, each dimension of the posterior variance was scaled up by a fixed factor (of 4 for MSA VAE, 1 for the AR-VAE model). 500 latent vectors were sampled

from the resulting scaled posterior distribution for each model. In the case of AR-VAE a further source of randomness was added to the autoregressive decoding process through temperature sampling (T = 0.5). The resulting sequences were separated into 6 approximately equally sized bins based on the number of differences from the input protein. 2 variants were chosen at random from each of these bins for synthesis, allowing variants having a range of distances to the original P19839 sequence to be tested.

**Conditional generation.** Mean and variance parameters for the posterior distribution over the latent variables were obtained by passing the one-hot encoded P19839 *luxA* sequence together with its one-hot encoded original solubility level (low) through the encoder. Sequence diversity was generated differently for the two types of model. For the MSA VAE, 100 latent vectors were sampled from the posterior for each of the two increased solubility levels (mid and high), and sequences were generated by passing the vectors together with the desired conditioning values through the decoder. For the AR-VAE models, the mean latent vector was used to generate all variants, with diversity amongst the 100 candidates generated for each conditioning level coming from temperature sampling (T = 0.3). 6 sequences were selected at random from each desired solubility level for each model. To prevent over-similarity amongst the synthesised sequences, members of pairs of selected sequences with less than 3 differences between them were replaced at random until all pairs satisfied this diversity criterion.

## Synthesis and cloning of sequence variants

The genes corresponding to the different variants of *luxA* were synthesized by Twist Bioscience. The variants were amplified from synthesized DNA fragments using primers F342/F343 (S4 File) and cloned under the control of the T7 promoter and upstream of C-terminal His-Tag sequence in plasmid pET28/16 [66] amplified with primers F340/F341 (S4 File) through Gibson assembly [67]. All plasmids were verified by Sanger sequencing. To study the function of the variants, the resulting plasmids were introduced in *E. coli* DH5 alpha carrying all the other genes of the *P. luminescens lux* operon on plasmid pDB283. This plasmid was obtained by deletion of *luxA* from plasmid pCM17 [68] by amplification using primers B731/LC545 and B732/LC327 (S4 File), followed by Gibson assembly of the two PCR fragments. Transformants were selected on LB agar containing kanamycin (50 μg/ml) and ampicillin (100 μg/ml). The plasmids described here are readily available from the authors upon request.

## Bioluminescence measurements

Strains were restreaked from −80˚C stocks on LB agar plates with kanamycin (50 $\mu$g/ml) and ampicillin (100 $\mu$g/ml). Pictures of the plates were taken with a C400 Azure Biosystem imager. The luminescence of 15 isolated colonies for each strain was measured. The acquisition time was adapted to the luminescence level. The reported relative luminescence units (RLUs) are normalized to the acquisition time used in each picture (S2 File).

## Protein solubility measurements

The recombinant plasmids were transformed into E. coli Rosetta cells. The transformants were grown in liquid Luria-Bertani medium containing chloramphenicol (20 $\mu$g/ml) plus ampicillin (100 $\mu$g/ml) at 30˚C until mid exponential phase (OD = 0.8-0.9). Isopropyl-$\beta$-D-thiogalacto-pyranoside (IPTG, 1 mM) was added to induce recombinant protein production and incubation was pursued for 3 h. Cells were resuspended in 1 ml of lysis buffer (Hepes 50 mM pH7.5, NaCl 0.4 mM, EDTA 1mM, DTT 1 mM, Triton X-100 0.5%, glycerol 10%), and then lysed on ice with a precellys homogenizer (Bertin Technologies) using the micro-organism lysing kit VK01 with the following conditions: 5 times for 30 s at 7800 rpm with 30 s of pause between

homogeneization steps. Soluble proteins were separated from aggregated proteins and cellular debris by centrifugation at 5000 g and 4˚C for 20 min. Pellets containing protein aggregates were resuspended in 1 ml of lysis buffer. For Western blot analysis, the samples were prepared in Laemmli buffer with addition of 10% beta-mercaptoethanol and denatured at 95˚C for 5 min. Soluble and insoluble fractions were run on a 4-12% Bis-Tris sodium dodecyl sulfate polyacrylamide gel electrophoresis (SDS-PAGE). Proteins were transferred to a nitrocellulose membrane (Invitrogen), which was blocked in 3% skim milk in PBS for 30 min and was successively incubated with primary (Anti-HisTag diluted 1:500 or Anti-GroEL diluted 1:1000 in blocking buffer) antibody and secondary antibody (diluted 1:10000) conjugated to DyLight 800 (Tebu), and detected under chemiluminescent imaging system (LI-COR Odyssey Instrument). The His tag was detected using the mouse monoclonal Anti-His-Tag antibody (Abcam). The GroEL control was detected with the mouse anti-GroEL monoclonal antibody (Abcam). Three washes for 5 min in PBS were performed after each incubation step. For dot blot analysis, 2% SDS and 10% $\beta$-mercaptoethanol were added to the samples before denaturing for 10 min at 95˚C. 5μl of the denatured samples were directly spotted on the nitrocellulose membrane and the antibody hybridization performed as for the Western blot. Protein levels were calculated using the Image Studio software package. Solubility data for the synthesised sequences is provided in S3 File.

### Variants excluded from analysis

In total 48 variants were selected for synthesis. Two variants were excluded from all analysis of experimental results: the first, one of the unconditional MSA VAE generations, was not synthesised successfully; the second (mc-1 5-9) showed insufficient expression levels to achieve reliable measurements (see S1 Fig).

## Supporting information

**S1 Fig. Quantitative analysis of solubility for each variant of *luxA* in comparison with wild type *luxA*.** A. LuxA tagged with a His-tag was quantified by western blot in the supernatant (S, soluble fraction) and in the pellet (P, insoluble fraction) compared to the empty vector pET28. GroEL was used as loading control. The His tag was detected using the mouse monoclonal Anti-His-Tag antibody, whereas the GroEL control was detected with the mouse anti-GroEL monoclonal antibody. The molecular mass is given in kilodaltons and indicated to the right of the membrane. Arrowheads indicate the position of recombinant proteins. The levels of solubility for variants of *luxA* generated by models trained on aligned (B) or raw (C) sequences were analysed by dot blotting. Aliquots of 5 μl of soluble (S) and insoluble (P) fractions from IPTG-induced Rosetta cells overexpressing variants of *luxA* were spotted on nitrocellulose membrane and their intensities were quantified using the Image Studio software package. The dot blots of 4 technical replicates used to compute solubility are shown.
(TIF)

**S2 Fig. Comparison of predicted solubility of sequences generated by conditioning vs natural sequences from training set by distance to *luxA* WT.** Mean and standard deviation of predicted solubility for 500 sequences generated by the conditional MSA VAE at the highest conditioning level, binned by distance to P19839 *luxA*, together with predicted solubility for all training sequences at equivalent distances (black crosses).
(TIF)

**S3 Fig. Predicted solubility of top variants generated by conditional and unconditional MSA VAE models across 10 random seeds.** To compare conditional and unconditional

models, we retrained 10 initialisations of both versions of the MSA VAE model using different random seeds. 500 variants were sampled from the posterior of the unconditional MSA VAE models and the highest level of the conditional models. For each random seed and each model, we computed the predicted solubility values of all sequences within 30 amino acid differences to *luxA* P19839. As a measure of the ability of the models to generate diverse high-solubility variants, we show the mean of the 50th highest predicted value across seeds. Error bars represent standard deviations. Three initialisations of the conditional model and three initialisations of the unconditional model generated insufficient variants within the distance threshold and were therefore excluded.
(TIF)

**S4 Fig. Performance of models retrained on subsets of the training data.** To assess the dependence of the models' ability to learn functional constraints from naturally occurring sequences on the number of training sequences, we created reduced training sets by randomly subsampling the full training set. For each reduced dataset size, three different random subsets of the training set of that size were sampled and used to retrain models. The *luxA* WT sequence was included in all training sets. After training, models were assessed by their ability to predict the luminescence of the synthesised variants, using the same metrics as in Table 1.
(TIF)

**S1 File. Generated sequences.**
(FA)

**S2 File. Luminescence measurements.**
(XLSX)

**S3 File. Solubility measurements and predicted solubility values for sequences generated by conditional models.**
(CSV)

**S4 File. Oligonucleotidue sequences.**
(FA)

**S1 Table. Amino acid reconstruction accuracies on heldout clusters by cluster sequence identity threshold used to construct train/test split.** To test generalisation to more distant family members, we retrained models using three different train/test splits, each of which was constructed by holding out clusters at a particular sequence identity threshold. In each case clusters were randomly added to the holdout set until the number of sequences in the holdout set was 20% of the total. Baseline VAE is a baseline model with the same architecture as MSA VAE, but the same latent dimension as AR-VAE, trained on raw sequence data.
(TIF)

## Author Contributions

**Conceptualization:** Alex Hawkins-Hooker, Sebastien Baur, David Bikard.

**Data curation:** Alex Hawkins-Hooker, Sebastien Baur.

**Formal analysis:** Alex Hawkins-Hooker.

**Investigation:** Alex Hawkins-Hooker, Florence Depardieu, Sebastien Baur, Guillaume Couairon, Arthur Chen, David Bikard.

**Methodology:** Alex Hawkins-Hooker, Florence Depardieu, Sebastien Baur, Guillaume Couairon, Arthur Chen, David Bikard.

**Project administration:** David Bikard.

**Resources:** David Bikard.

**Software:** Alex Hawkins-Hooker, Sebastien Baur, Guillaume Couairon, Arthur Chen.

**Supervision:** David Bikard.

**Validation:** Alex Hawkins-Hooker.

**Visualization:** Alex Hawkins-Hooker, David Bikard.

**Writing – original draft:** Alex Hawkins-Hooker, David Bikard.

**Writing – review & editing:** Alex Hawkins-Hooker, David Bikard.

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
