## [Decision Letter · Decision Letter 0]

18 Oct 2020

Dear Bikard,

Thank you very much for submitting your manuscript "Generating functional protein variants with variational autoencoders" for consideration at PLOS Computational Biology. As with all papers reviewed by the journal, your manuscript was reviewed by members of the editorial board and by several independent reviewers. The reviewers appreciated the attention to an important topic. Based on the reviews, we are likely to accept this manuscript for publication, providing that you modify the manuscript according to the review recommendations.

 In particular there is a need to address remaining concerns around controlling the solubility level and  baseline comparisons.

Sincerely,

Christine A. Orengo

Associate Editor

PLOS Computational Biology

Arne Elofsson

Deputy Editor

PLOS Computational Biology

[LINK]

Reviewer's Responses to Questions

**Comments to the Authors:**

Reviewer #1: The authors addressed reviewers’ comments thoroughly. The revised version is improved in quality. I have no further suggestions to make.

Reviewer #2: The paper presents a study of generating sequences for luciferase domains based on training VAEs from a set of 70,000 protein sequences similar to luciferase sequences. Some sequences from each model are realised and their activity measured.

The generated sequences mostly do luminesce, though none as strongly as WT. [It would be good to know about more controls than just the one -are any of the 70k sequences better than the measured WT?)

The need to do this (given there are 70,000 examples) is not strongly motivated in itself, but the paper did go on to show conditional versions where solubility of the generated sequences is controlled, while retaining function, though it’s not shown whether this is better or worse than just directly modifying the WT to adjust solubility in ways that are well-known (especially given the extensive MSA - cf PROSS).

I find that the authors in both the paper and the response to previous reviewers’ comments do seem to consider “edit distance” a bit naively. Yes, 5 random mutations [uniform in position and across AA type] are very likely to destroy function, but it would not be hard to choose sensible mutations that are unlikely to destroy function, especially given such an extensive MSA.

I agree with the previous reviewers (and the authors) that there is a lot of work in this direction, and this is very limited by needing such a large dataset of similar proteins whose function we're just recapitulating. I do nevertheless find it worthy enough for publication. I find the contrast between the “MSA” and AR versions of the model interesting.

Reviewer #3: Overall, the authors adequately addressed the issues raised in the initial review and present a reasonable plan for a re-submission. There are, however, some remaining points that should be addressed:

To demonstrate the value of conditioning on solubility, and support the claim that the conditional model can be used to “control the solubility level while introducing relatively few mutations”, the authors show a plot which compares the predicted solubility of sequences generated by the conditional model and sequences in the training set. To better support for these claims, the authors should also include the predicted solubility of sequences generated by the model without conditioning on solubility. This would more directly address the value of using a conditional model.

To address a concern (raised by multiple reviewers) that the model should be more extensively compared to baselines, the authors present some additional analysis, where they show that the model ELBOs show a higher correlation with protein luminescence than BLOSUM or HMM scores. The results shown provide compelling evidence that the MSA VAE can generate novel variants that maintain luciferase activity. The HMM does, however, reasonably predict the luminescence of AR-VAE generated sequences, suggesting that one could potentially generate functional proteins by sampling from the HMM.

A final, minor, point is that the authors indicate that the sequences generated using the conditional model are more similar to the wild type sequence than sequences generated by the VAE without conditioning on solubility. Aggregate statistics for their luciferase assays are nonetheless reported in the abstract. This is misleading and is an overestimate of the performance of the non-conditional model. The number of functional variants that are far away from training sequences is less than the presented amount.

**Have all data underlying the figures and results presented in the manuscript been provided?**

Reviewer #1: Yes

Reviewer #2: **No: **While code and network weights are provided, with a license, I did not find a pointer to the associated data.

I would expect to see the 3000 sequences generated from each model, the predicted solubiliities, as well as actual luminescence and solubility for each of the measured proteins.

Reviewer #3: None

PLOS authors have the option to publish the peer review history of their article (what does this mean?). If published, this will include your full peer review and any attached files.

Reviewer #1: No

Reviewer #2: No

Reviewer #3: No
---

## [Decision Letter · Decision Letter 1]

25 Jan 2021

Dear Dr Bikard,

We are pleased to inform you that your manuscript 'Generating functional protein variants with variational autoencoders' has been provisionally accepted for publication in PLOS Computational Biology.

Best regards,

Christine A. Orengo

Associate Editor

PLOS Computational Biology

Arne Elofsson

Deputy Editor

PLOS Computational Biology

Reviewer's Responses to Questions

**Comments to the Authors:**

Reviewer #3: The authors addressed all my comments.

**Have all data underlying the figures and results presented in the manuscript been provided?**

Reviewer #3: Yes

PLOS authors have the option to publish the peer review history of their article (what does this mean?). If published, this will include your full peer review and any attached files.

Reviewer #3: No

---

## [Editor Report · Acceptance letter]

22 Feb 2021

PCOMPBIOL-D-20-01497R1 

Generating functional protein variants with variational autoencoders

Dear Dr Bikard,

I am pleased to inform you that your manuscript has been formally accepted for publication in PLOS Computational Biology. Your manuscript is now with our production department and you will be notified of the publication date in due course.

With kind regards,

Alice Ellingham
